Time arrow in published clinical studies/trials indexed in MEDLINE: a systematic analysis of retrospective vs. prospective study design, from 1960 to 2017

Ciulla Michele M. michele.ciulla@unimi.it 1
Vivona Patrizia 2
1 Laboratory of Clinical Informatics and Cardiovascular Imaging; Department of Clinical Sciences and Community Health, University of Milan , Milan , Italy
2 Cardiovascular Diseases Unit, Fondazione IRCCS Cà Granda Ospedale Maggiore Policlinico , Milan , Italy
Mastrolia Salvatore Andrea
Electronic publication date: 2019 Feb 1
Publication date: 2019
Volume: 7
Electronic Location ID: e6363
Received 2018 Mar 13; Accepted 2018 Dec 30
Copyright: ©2019 Ciulla and Vivona
Copyright year: 2019
Copyright holder: Ciulla and Vivona
License: This is an open access article distributed under the terms of the Creative Commons Attribution License, which permits unrestricted use, distribution, reproduction and adaptation in any medium and for any purpose provided that it is properly attributed. For attribution, the original author(s), title, publication source (PeerJ) and either DOI or URL of the article must be cited.
License URL: https://creativecommons.org/licenses/by/4.0/

Keywords: Biological life, Clinical trials, Chronobiology, Clinical study, Disease, Retrospective study, Time variable, Prospective study, MedLine, Aging, Demographic data, Bibliometric study

Funding: The authors received no funding for this work.

==============================
Clinical studies/trials are experiments or observations on human subjects considered by the scientific community the most appropriate instrument to answer specific research questions on interventions on health outcomes. The time-line of the observations might be focused on a single time point or to follow time, backward or forward, in the so called, respectively, retrospective and prospective study design. Since the retrospective approach has been criticized for the possible sources of errors due to bias and confounding, we aimed this study to assess if there is a prevalence of retrospective vs. prospective design in the clinical studies/trials by querying MEDLINE. Our results on a sample of 1,438,872 studies/trials, (yrs 1960–2017), support a prevalence of retrospective, respectively 55% vs. 45%. To explain this result, a random sub-sample of studies where the country of origin was reported (n = 1,576) was categorized in high and low-income based onthe nominal Gross Domestic Product (GDP) and matched with the topic of the research. As expected, the absolute majority of studies/trials are carried on by high-income countries, respectively 86% vs. 14%; even if a slight prevalence of retrospective was recorded in both income groups, for the most part prospective studies are carried out by high-GDP countries, 85% vs. 15%. Finally, the differences in the design of the study are understandable when considering the topic of the research.

Introduction and Aims

We generally consider time as a quantity that in clinical studies/trials plays the role of independent variable having, often, a real value on an axis of a graph; but time is just a way to give order to changes and, in biology, time is essentially rhythm of changes, with short and long cycles that are controlled by specific allocated resources that are the matter of chronobiology (Goldbeter, 1996). Usually clinical studies/trials are focused on diseases that are a common cause of chronodisruption at multiple levels (Oldham, Lee & Desan, 2016), meaning that circadian rhythms are severely altered among the patients, depending on the severity of the disease; this disruption expands from the cellular level, where the genes that regulate the biological clock are expressed, up to the subjective perspective of the patient including pain. These studies are generally aimed at identifying causes of disease and to set up/verify possible strategies to prevent or care them (Friedman, Furberg & DeMets, 2010). Usually the observation of the variable(s) under study is fixed on singular time points or is followed backward or forward in the so called, respectively, retrospective and prospective design of the study. In such view the availability of comprehensive clinical databases, e.g., compiled during the hospitalization, allows the clinical researcher to force easily backward the arrow of time in search of possible causal links. This approach, that is undoubtedly cheaper, has often been criticized (Secrier & Schneider, 2014) for the possible sources of errors due to bias and confounding, that are, indeed, more common if compared with prospective studies (Mantel & Haenszel, 1959).

In order to evaluate the tendency of researchers to use a retrospective design in studies/trials by turning backward the “arrow of time”, in this study we firstly searched MEDLINE and obtained the number of retrospective and prospective matching studies, starting from 1960 to 31 December 2017, an interval that was considered representative since, before 1960, very few studies are available. Secondly, in order to speculate on the reasons of this choice, we randomly choose a sub-sample of studies, where the country of origin of the first author was clearly stated, that was matched with the nominal Gross Domestic Product (GDP) obtained from the World Economic Outlook (WEO) by the International Monetary Fund (IMF) (World Economic and Financial Surveys, World Economic Outlook Database, 2017). Thirdly, to evaluate the prevalence of retrospective versus prospective studies in the different thematic areas, the results have been grouped accordingly. Finally, in the discussion the conclusions are matched with the opinions previously gathered through a random sample of researchers, matching as authors of the papers retrieved, that were obtained by e-mailing, by using a published survey questionnaire (Survey Monkeys, 2018).

Methods

The MEDLINE database was queried through PubMed, the public access web portal of the National Library of Medicine, National Institutes of Health-USA, in September 2017 by using the keywords “retrospective study” and “prospective study”; these words were assumed as reference to describe the two types of approach. For the purpose of this research, after evaluating/comparing the possible strategies, including a comparison with other databases such as EMBASE, (see also limitations of the study), in agreement with our reference librarian, we relied on free text search by using the “best match”, a function available on PubMed (Fiorini et al., 2018) that includes MESH. The search string used was: retrospective study AND“1960/01/01”[PDat]: “2017/12/31”[PDat]

prospective study AND“1960/01/01”[PDat]: “2017/12/31”[PDat].

For the general statistics, even if few matching records were found starting from 1913, the “publication date” of the search query was limited by starting from 1960, where at least 10 matching documents for both category were found. For the assessment of the country of origin and topic, we randomly sampled a minimum of five studies per decade where the country of origin of the first author was clearly stated and the topic was assessable by the availability of a full text. Since in most cases it was not possible to find all the information needed, this sample was built starting from 2004 and placed in a comprehensive database including the over-mentioned variables.

Finally, we created two GDP-based macro-categories that permitted to distinguish high-income countries of origin of the research from low ones; the cut-off for incomes was >467,775 millions of US $; this arbitrary value, corresponding to 75% over the mean, allowed to include a larger number of countries into the high-GDP with respect to other criteria such as the G-7. Thus, the two GDP categories were matched with the topic of the studies.

MMC and PV defined the search strategy, in agreement with our reference librarian (see acknowledgments section), PV performed the search strategy. All the disagreements were solved by carefully checking the search strategy previously.

Results

The results support that, since the sixties, there are significant numerical differences with a majority of retrospective studies. Out of a 1,438,872 matching records, 787,938 (55%) were retrospective whereas 650,934 (45%) prospective. By examining the decades, this trend began in 1960 and, except for the decade 1980–1990, is maintained and widen considerably in the last years. Indeed, in the last 5 years 278,354 (58.6%) retrospective vs. 196,679 (41.4%) prospective studies were found, corresponding to a difference of +81,675 retrospective studies (Fig. 1). Furthermore, as stated in the methods section, even if the search was limited to the interval 1960–2017, a difference is appreciable since the starting coverage of the database; the keywords “retrospective study” appears for the first time in 1913 while “prospective study” in 1923, with only one matching document for each type of study. This difference is also reproduced in the sub-sample (years 2004–2015) made of 1,576 matching records, of which 903 (57%) were retrospective whereas 673 (43%) prospective. The general prevalence of studies attributable to the high-GDP countries was very high accounting for 86% of the total vs. 14%; this data support that conception, realization of a clinical study/trial and access to publication is related to the GDP (Fig. 2). Furthermore the 85% of prospective studies are ascribable to the high-GDP group while only a 15% to the low-GDP, and this even if a prevalence of retrospective studies was recorded in the high-GDP. Finally these differences are understandable when considering the topic of the research that requires a specific design (Fig. 3).

Figure 1 Number of matching documents retrieved from MedLine years 1960–2017 by using the keywords indicated in the method section.

(A) Area graph showing the number of matching documents retrieved from MedLine years 1960–2017 by using the keywords indicated in the method section. On the x axis are reported the decades starting from 1960, on the y axis, the absolute number of published papers. (B) Pie chart reporting the percentage of retrospective and prospective studies.

Figure 2 Pie chart showing the prevalence of Retrospective (A) vs. Prospective studies (B) in high and low-GDP countries as retrieved from MEDLINE.

Figure 3 Prevalence of the topic in the retrospective and prospective studies retrieved from MedLine.

Bar chart showing the prevalence of the topic in the retrospective (A) and prospective studies (B) as retrieved from MedLine.

Discussion

Time is the most intriguing variable in the universe and, on earth, it affects irreversibly biological life since every form of life has a non-zero probability of dying that increases with the flow of time in a general process known as biological aging. Since it is impossible to reverse the course of events, living organisms must adapt reproduction, development and evolution to cope time and its rhythms by committing specific biological resources that are the matter of chronobiology, making time a measure of the cyclical and stochastic changes in biology rather than an absolute physical quantity. Biological life with aging is invariably associated with the experience of disease, whose incidence seems to be ruled by chance (Ciulla, 2015b) but when clinical researchers study diseases, generally, to preserve human life and avoid unnecessary suffering, what they actually do of the time variable, namely, how the variables are followed, or sampled, in time, backward or forward? Indeed any study on diseases might takes into account the “point of view” which might suggest misleading cause–effect relationship (Ciulla, 2015a). Even if the our objective was not so ambitious to dismantle the mechanism of time in diseases, in the present study, based on a MEDLINE search strategy starting from 1960, we highlighted that, when studying disease, the preferred choice of the study design is the retrospective one and this is nontrivial as it allows us to make some considerations. We can assume that the retrospective study design is prevalent in the scientific literature possibly since it represent: (1) a cheaper approach to make forecasting on disease development, (2) an easy choice because of the availability of clinical records or, simply, (3) a wrong choice due to lack of statistical knowledge. These assumption are corroborated by the opinions of a random sample of researchers, matching as authors of the papers retrieved, that were obtained by e-mailing, previously, by using a published survey questionnaire (Survey Monkeys, 2018). In particular, in choosing the retrospective design, the involved researchers feel that they have been limited by the availability of funds and instead favored by the easy access to a clinical database. Furthermore, the surveyed researchers believe that the prospective design is more expensive, statistically appropriate and clinical predictable. At this regard, it should be said that there is no direct evidence of the inferiority of retrospective studies if well conducted (Nagurney et al., 2005); however, retrospective studies are limited by the level of knowledge and availability of data at the time of collection, on the contrary, prospective studies may, theoretically, incorporate any possible newer variable. These opinions are supported by our results since the absolute majority of studies/trials are almost always conducted by high-GDP countries and the 85% of prospective studies are ascribable to the high-GDP group while only a 15% to the low-GDP. A further key can be found in the topic of the studies/trials where most of prospective are associated with “cardiovascular”, a topic known to catalyze great investments from the pharmaceutical companies, and “surgery”, which benefits from a short-term prospective outcome.

Conclusions

In clinical studies/trials the time variable is fundamental in the design the study; by analyzing MEDLINE database 1960–2017, we show that: 1—most of clinical studies/trial use a retrospective design of the study even if there is no direct evidence of its superiority if compared to the prospective one, on the other hand, there are no evidence of an intrinsic inferiority of the retrospective design in statistical precision and in clinical predictability (Nagurney et al., 2005); 2—most of prospective studies were conducted by high-GDP countries supporting that the cost is, as expected, one of the main constraints and this possibly concerns the cost of research, as also highlighted by other studies, of some years ago, on imbalance in health research resources (Barreto Mauricio, 2009) and on the consequences of globalization of clinical research (Glickman et al., 2009); and 3—some research topics are more frequently associated with the perspective design and this may be linked, possibly, to the availability of funds and the need of a specific short-term design of the study.

Limitations

This study has several limitations that should be acknowledged, Firstly, the key words retrospective and prospective are assumed as reference to describe two approaches in setting up a clinical study/research and this may results in some imprecisions due to a broad use of these terms; in this regard, the long lasting use of this terminology confirms their established semantic use. Secondly, bibliometric methods examine a body of work at a macro level and our results must be interpreted in light of this. Using a simple search has the benefits of being readily understood and easily replicated but inevitably produces some imprecision or noise. In our study 93% of prospective studies and 95% of retrospective studies were indexed as human, some were index as animal studies, some were indexed as both human and animal studies, and some were indexed as neither. In any case, these studies have been classified this way according to MESH and, therefore, included in the results of the research which was conducted with a correct syntax. Also, about 8% of prospective studies were also retrieved by the search for retrospective studies and are included in both analyses; thirdly, MEDLINE is constantly expanding thus our study represents a partial view over a rather long time interval. Finally, by focusing on the first author, we emphasized the authorship by losing the opportunity to highlight international collaborations.

Supplemental Information

Supplemental Information 1 Revised search strategy and contribution

Click here for additional data file.

Supplemental Information 2 Raw data

Partial data collection.

Click here for additional data file.

We acknowledge Elena Bernardini, Head Librarian, Biomedical Libraries, University of Milan, Italy.

Additional Information and Declarations

Competing Interests

Author Contributions

Data Availability

The authors declare there are no competing interests.

Michele M. Ciulla conceived and designed the experiments, performed the experiments, analyzed the data, contributed reagents/materials/analysis tools, prepared figures and/or tables, authored or reviewed drafts of the paper, approved the final draft.

Patrizia Vivona performed the experiments, analyzed the data, contributed reagents/materials/analysis tools, authored or reviewed drafts of the paper, approved the final draft.

The following information was supplied regarding data availability:

The raw data are available in the Supplemental File.

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
