# Peer review of "Time arrow in published clinical studies/trials indexed in MEDLINE: a systematic analysis of retrospective vs. prospective study design, from 1960 to 2017"

_PeerJ, doi:10.7717/peerj.6363_

## Round 0.1 · original submission · Major Revisions

Dear Authors,

The Reviewers found your manuscript very interesting, moreover recommending a thorough revision in order to achieve publication.
I would suggest to take into consideration the Reviewers' comments, discuss and incorporate them within your manuscript in order to reach the standard requested for publication.

I would suggest the authors to pay special attention to the improvement of written English. Submitting this work to a proof reading professional service might be a good option in order to achieve improvement of the language resulting in a better readability of the manuscript.
Best regards

Salvatore Andrea Mastrolia
PeerJ Academic Editor

·

Basic reporting

1.1. Clear and unambiguous, professional English used throughout.
The article would require considerabe editing to get to the clarity and simplicity of expression needed.

1.2 Literature references, sufficient field background/context provided.
More complete review of the existing literature would be helpful. There is some existing literature on distribution of research in narrower topic areas that could be interesting to discuss, here are some exampes
Falagas ME, Papastamataki PA, Bliziotis IA. A bibliometric analysis of research productivity in Parasitology by different world regions during a 9-year period (1995–2003). BMC Infect Dis. BioMed Central; 2006 Dec 17;6(1):56.

Falagas ME, Karavasiou AI, Bliziotis IA. A bibliometric analysis of global trends of research productivity in tropical medicine. Acta Trop. Elsevier; 2006 Oct 1;99(2–3):155–9.
Michalopoulos A, Falagas ME. A Bibliometric Analysis of Global Research Production in Respiratory Medicine. Chest. Elsevier; 2005 Dec 1;128(6):3993–8.


1.3 Professional article structure, figs, tables. Raw data shared.
The artice is well organized and adhers to the IMRAD structure.
Figures need to be reconsidered. Results that are simple enough to be displayed in pie charts should usually just be presented numerically in the text or, when there are many categories, in a table. I do not think the pie charts are needed. The line graph should not be shade.

In Figure 1, top panel, you have graphed 10 year intervals until the final, which is 5 years. This erroneously looks like production has declined.

1.4 Self-contained with relevant results to hypotheses.
Good

Experimental design

2.1 Original primary research within Aims and Scope of the journal.
No major issues

2.2 Rigorous investigation performed to a high technical & ethical standard.
The reporting of the search makes it difficult to assess but 1. You must report your search strategy in greater detail. I cannot replicate it or determine if it is of good quality. The following should be reported:
The date you searched
The interface or platform used to search MEDLINE (this could be Ovid, Ebscohost, PubMed, to name a few of the possibilities)
A transcript of the search


Both retrospective studies and prospective studies exist as MeSH terms and each has narrower terms representing specific study designs. The limit to humans is reasonable. I do not know of any article type limit for "original study". Please clarify this.
.

2.3 Methods described with sufficient detail & information to replicate.
There are quite a few points here for the authors to address.
I attempted to replicate your search, although I had to make some assumptions. My results are below. I would expect to see a transcript like this in the appendix.

Ovid MEDLINE, searched April 13, 2018.
Searches
1 exp Retrospective Studies/ 681063
2 exp Prospective Studies/ 469502
3 limit 2 to (yr="1960 - 2015" and journal article) 411859
4 limit 1 to (yr="1960 - 2015" and journal article) 579375
5 1 or 2 1131628

These proportions work out a little differently than from your search; 58% retrospective and 42% prospective.

Please describe your process of randomly sampling studies.

Why was it necessary to have the full text and what do you mean "the topic was assessable by the availability of a full text? Does this mean a free full text was linked in PubMed, or that the journal was available in your institutional library or that you requested articles through interlibrary loan. This could seriously bias your findings. Wikipedia describes this in an entry for "FUTON" bias (https://en.wikipedia.org/wiki/FUTON_bias)

How did you determine country of origin? What is the country where the research was performed or the country listed for the corresponding author? How did you classify studies that either recruited patients from multiple countries or had authors from multiple countries (although this would only be a factor if there were a mix of high and low income countries represented).

You classify the research into categores – Aesthetics, ophamlogy, pneumology etc. What is the typology you are using and how did you determine, from the article, where is should be classified? You may be describing this in lines 110-111, but much more detail would be needed.

Please provide a source for the GDP figures used to classify countries as high or low income. Do I understand correctly that all of the countries in the world were classified and 25% fell in the high income range and 75% in the low income range?

I am not confident about which results pertain to the period 1960-2015 and which results pertain to 2004-2015,

Validity of the findings

Without more information on the methods, it is difficut to assess the accurracy of the results from MEDLINE however, the numbers presented seem reasonable.

An issue that needs to be considered is whether research conducted by investigators living and working in low income countries is published in MEDLINE indexed journals? See, for example Salager-Meyer F. Scientific publishing in developing countries: Challenges for the future. J English Acad Purp. Pergamon; 2008 Apr 1;7(2):121–32. One hypothesis about the rise of "predatory publishers" is that they provide access
3.1 Data is robust, statistically sound, & controlled
The data file made available with the manuscript would require additional annotation or would need to be organized differently before being interpretable to another investigator.

·

Basic reporting

1) The introduction and background do not fully describe the field being reviewed. No research question is given. From the abstract, an objective is alluded to, but not fully explained: line 45-46 we aimed this study to assess if there is a prevalence of retrospective vs prospective design in the clinical studies/trials by querying MedLine. Considering all research as either retrospective or prospective without considering quality seems simplistic when there are many observational designs that may be appropriate to the research question.

2) Authors do not describe a clear need for the review. The intended audience (i.e. which field) is not described. The relevance to readers of PeerJ in both the immediate field and any associated fields is not clear.


3) The English language needs improvement to ensure that the audience can clearly understand your text. Some examples where the language could be improved include:

Lines 39-43 Clarity at the outset is very important to orient the reader. These sentences are too long and complicated: "Clinical studies/trial are considered by the scientific community the most appropriate instrument to identify causes of disease and/or to set up/verify possible strategies to prevent or care them. In such approach the attention to the time-line of the variables under study might be focused on a single time point or to follow time, backward or forward, in the so called, respectively, retrospective and prospective design."

Lines 74-76 Very unclear. Consider adding some definition of the concepts “time”, “rhythm of changes”, “chronobiology”. Also the sentence is too long.
"We generally consider time as a quantity that in clinical studies/trials plays the role of independent variable having, often, a real value on an axis of a graph; but time is just a way to give order to changes and, in biology, time is essentially rhythm of changes, with short and long cycles that are controlled by specific allocated resources that are the matter of chronobiology."

Line 150-151 Very unclear – are the authors likening publication and research trends to human biology? I do not understand.
"Biological aging is, invariably, accompanied by the experience of disease, a domain that seems to be ruled by chance but when clinical researchers study diseases, generally, to preserve human life and avoid unnecessary suffering, what they actually do of the time variable, namely, how the variables are followed, or sampled, in time, backward or forward?"

Line 180 this phrase is not correct English: "Clinical studies/trial the time variable is fundamental in the design the study;"

4a) Citations:
Some citations given seem relevant but not all assertions by the authors are backed up by citations for example:

81-83 "Usually the observation of the variable(s) under study is fixed on singular time points or is followed backward or forward in the so called, respectively, retrospective and prospective design of the study."
I think a better overview of observational studies should be found and cited.

4b) This assertion is incorrect, and based on a 1959 citation:
Line 86-87 - sources of errors due to bias and confounding, that are indeed, more common if compared with prospective studies."

Research methods have evolved since 1959 and rather than labelling a type of research design “biased” it is more appropriate to discuss how well the research design suits the research question. A case-control, or other retrospective study design may be very appropriate, for example, in the study of rare diseases. “For rare diseases or the diseases that have a long latency period between an exposure and outcome, case control studies are often the only feasible choice.” [See: https://www.ncbi.nlm.nih.gov/pmc/articles/PMC3326852/]

5) The structure conforms to PeerJ standards with the appropriate headings given.

6) Figures are well labelled and described.

7) Raw data has been supplied.

Experimental design

8) Research question is not provided. An identified knowledge gap is not described, nor how this research fills such a gap.

9) Methods are not described with sufficient detail and information to replicate.
The full, original search strategy should be given, including which platform was used to search MEDLINE. The reader also needs to see how the limits of “Human” and years were applied in greater detail.

10) It is unclear how authors could tell whether the studies were actually retrospective or prospective with a yield of over 1 million records.

11) There is no discussion of how technology may affect the quality or bias of retrospective or prospective studies. Very large datasets are available to researchers, and the types of large registry studies we see today would not have been possible then. I find the inclusion and comparison of pre-computer research and research designs inappropriate, and perhaps this should have been addressed in the limitations. Indeed, the very reason the authors chose this time interval was because (line 90-92) “starting from 1960 to 2015, an interval that was considered representative since, before 1960, very few studies are available.” They were are not available because they have not been added to a very large modern database.

Validity of the findings

no comments

Additional comments

I hope the authors consider adding more people to their team. I think the work would benefit from enlisting a professional librarian with experience and advanced skills in searching the MEDLINE database, and clearly reporting the search to a current standard (see: https://www.cadth.ca/resources/finding-evidence/press).

The manuscript would benefit from the involvement of someone with strong English writing skills to the team who can clearly articulate their work for the audience.

Some other minor comments/revisions:

Line 76-79 – the example given about “chronobiology” and illnesses of “chronodisruption” is unclear. Was the study limited to these types of studies? Please explain the relevance.

Line 85 – spelling error: cheeper should be cheaper

Throughout: MedLine should be written as MEDLINE

Line 195-196 – Please remove this line: “Secondly, a paper describing an off-label use does not necessarily support a possible future clinical use” – unless it can somehow be shown to be relevant to the objectives of the study.

---

## Round 0.2 · Major Revisions

Dear Authors,
I very much appreciated the efforts provided in addressing the Reviewer's comments. However, the manuscript still presents some important criticism that at the moment do not allow it to be accepted for publication in PeerJ.

Due to the interest of the subject, I have decided to give an additional round for comments but I strongly encourage the Authors to take into consideration the Reviewer's comments, discuss, and incorporate them within the manuscript in order to reach the standard requested for publication.

Best regards

Salvatore Andrea Mastrolia
PeerJ Academic Editor

·

Basic reporting

The introduction needs to include the research question or problem, in one simple sentence. Also please clearly state the aims of the paper. (Explain in one aim per sentence).

The first paragraph contains excessive background information which is difficult (impossible?) to understand without knowing its relevance to the research question. Lines 90-98 are actually methods, and they do not belong in the "Introduction and Aims"

Clarity of the written English could be improved throughout the paper.

Experimental design

The research question is not well defined. See comment above. The reader should understand the research problem in the first paragraph.

The search methods are still not reported to a current expected standard.
At the minimum, the authors should provide the exact search string that was used to search Pubmed, including boolean operators and date limits.
This can be provided either by cutting and pasting, or screen shot either in the body of the paper, or an appendix. If more than one search was done, all the search strings should be reported. Without this information, there is not sufficient detail & information to replicate.

Validity of the findings

no comments

Additional comments

I thank the authors for the improvements to this manuscript, and for their letter explaining the actions taken to address my comments.

I strongly urge the authors to add more people to their team. The paper needs to be edited by someone with very strong English writing skills. A professional librarian could ensure that the MEDLINE (actually Pubmed) search is executed and reported accurately.

---

## Round 0.3 · Minor Revisions

Dear Authors,

The Reviewers are favorable to the publication of your manuscript in PeerJ after a minor revision.

Please incorporate or discuss the suggested changes and submit a revised version of your manuscript in order to achieve publication .
Best regards

Salvatore Andrea Mastrolia
PeerJ Academic Editor

·

Basic reporting

Thank you for the opportunity to review this manuscript in its revised form. I agree that the authors have robustly assessed alternative search strategies and selected one that balances simplicity and accuracy. This is a sound approach for a bibliometric study such as this.
The English usage and style in this version is much easier to read and understand. I will leave the decision as to whether further refinement is needed to the journal editors.
There are a few remaining issues, I think the authors will be easily able to address all of them.
Finally, may I say that this paper would have been much stronger if the authors had examined the prior literature on research capacity and productivity across the spectrum of low to high resource countries.

Experimental design

Issue 1. Reporting of the search strategy. You need to say what you did, exactly, so others can replicate your search.
Please change the reporting in methods, line 111-113, reporting this almost exactly as you did in the reply to reviewers document:
We searched PubMed in September 2017 as follows;
(retrospective study AND ( "1960/01/01"[PDat] : "2017/12/31"[PDat] )) sorted by Best Match
(prospective study AND ( "1960/01/01"[PDat] : "2017/12/31"[PDat] )), sorted by Best Match

When I paste the string
(retrospective study AND ( "1960/01/01"[PDat] : "2017/12/31"[PDat] )) into PubMed today, October 18, 2018, I retrieve 744,511 records, pretty close to your reported 787,938. When I enter what you have in methods “retrospective study” using best match sort and keeping the publication dates the same above, I get 115,502 records, which is vastly different.
Issue 2 – Survey findings

Lines 177 – 182. Now you are citing the actual survey instrument in SurveyMonkey. In earlier version of the manuscript, you cited this preprint in ResearchGate: Ciulla MM. The arrow of time in clinical studies. Retrospective vs prospective: a survey on the propensity of researchers in clinical study’s design. ResearchGate (2016). doi:10.13140/RG.2.1.1725.1443

The citation of the preprint is necessary unless you want to include the methods and results in this manuscript. Either way, a couple of points are; 1. what is the response rate of the set you were able to survey? 2. What were the reasons investigators chose a prospective design? The pre-print seems to show results only for those who chose a retrospective design. 3. You state in line 181-182 “Furthermore, the surveyed researchers believe that the prospective design is more expensive, statistically accurate and clinical predictable.” The survey question reads “statistically appropriate” so the wording in the manuscript should also say “appropriate”.


Issue 3

In the Discussion, line 213 you state “previously ascertained through a comparison also with other databases such as EMBASE” 
 – introducing Embase here is too late, assuming you mean the exploratory work you did in Embase and described in the reply to reviewers document. You should report this work in the methods section, lines 114-116, pretty much the way you described it in the reply to reviewers.

Validity of the findings

An assumption that the authors seem to be making throughout the paper is that all articles indexed in PubMed are clinical studies. Many won’t be, they may be animal studies, studies of health systems, even methodological research on retrospective or prospective study designs. A note in Limitations section could just indicate you are using this term very broadly and there will be some imprecision in your results.

Another limitation to mention is that only the address of the corresponding author was considered (and this is a limitation of MEDLINE) whereas other research databases such as Web of Science provide all authors, allowing consideration of international collaborations (see for example Huffman MD, Baldridge A, Bloomfield GS, Colantonio LD, Prabhakaran P, Ajay VS,Suh S, Lewison G, Prabhakaran D. Global cardiovascular research output, citations, and collaborations: a time-trend, bibliometric analysis (1999-2008).PLoS One. 2013 Dec 31;8(12):e83440. )

Additional comments

Key word selection: I would add “bibliometric study” to represent the methods used.

Line 92 – delete the word appropriately

Line 116 – I am not convinced at all that it prevents classification problems. I recommend deleting this phrase. As well citation 8 (Volpato) is not a great source for this, as the best match was deployed in 2017 and Volpato does not test anything like it. I would defer to the National Library of Medicine documentation or an article that studies Best Match directly such as Fiorini N, Canese K, Starchenko G, Kireev E, Kim W, Miller V, Osipov M,Kholodov M, Ismagilov R, Mohan S, Ostell J, Lu Z. Best Match: New relevance search for PubMed. PLoS Biol. 2018 Aug 28;16(8):e2005343. doi:10.1371/journal.pbio.2005343. eCollection 2018 Aug. PubMed PMID: 30153250; PubMed Central PMCID: PMC6112631.

Line 165. Reference 8 (Volpato) is cited here as well. Likely, 9 is the intended reference (Ciulla)

---

## Round 0.4 · Minor Revisions

Dear Authors,

Reviewer 1 provided their comments and is again favorable to the publication of your manuscript in PeerJ after a minor revision.

Moreover, I need to ask you an additional effort in order to address all comments within the next revision of your manuscript, since this will be the fifth version.

It is my intention to give a final approval or rejection of your manuscript after the submission of the next version.

Please incorporate or discuss the suggested changes and submit a revised version of your manuscript in order to achieve publication.

Best regards

Salvatore Andrea Mastrolia
PeerJ Academic Editor

·

Basic reporting

In terms of the overall structure of the document, the quality of the English writing is now ranges from good to excellent. I did notice a little type in line 222 "MEDLINE is t is constantly".

I think the limitations section (line 212 – 224) belongs in the discussion section rather than the conclusion.

Experimental design

First, the authors (or their advisors) do not seem aware that PubMed Best Match is a ranking algorithm. It has a very small effect on which articles are retrieved, but has a significant effect on the order they are presented it. Thus it doesn't address bias and it doesn't exclude animal studies. This is pretty easy to test empirically and I suggest you repeat your searches comparing counts for best match and the same search sorted by date. Then limit each to human, than to animal and repeat the sorts. I think you will quickly understand what is happening. Therefore, line 117-118 needs to be altered from "we relied on the free text search by using the "best match", a function available on PubMed that prevents incorrect classification problems by including MESH." to a simple statement that they searched using the Best Match feature, without making any claims about what that feature does.

Likewise, under the heading "limitations" in lines 218 where the authors state "Secondly, by using the keywords without the quotes, inevitably, produces some noise, nonetheless, by using the recently updated “best match” function for PubMed that includes MESH, we prevented most incorrect classification problems." This is simply false and the authors need to accept that their methods have some limitations. May I suggest something along these lines:

"Secondly, bibliometric methods examine a body of work at a macro level and our results must be interpreted in light of this. Using a simple search has the benefits of being readily understood and easily replicated but inevitably produces some imprecision or noise. In our study 93% of prospective studies and 95% of retrospective studies were indexed as human, some were index as animal studies, some were indexed as both human and animal studies, and some were indexed as neither. As well, about 8% of prospective studies were also retrieved by the search for retrospective studies and are included in both analyses. "

Some data;
Total Animal % Human % Both Human and Animal % Neither %
prospective 652002 21520 3.3 607159 93 10414 1.6 33737 5
retrospective 789806 18207 2.3 746473 95 7746 0.9 32872 4

both pro & retro 53429
% of prospective 8.1%

Validity of the findings

The authors may have misunderstood my suggestion that using the address of only the first author may overlook international collaborations between researchers in high and low GDP countries. This leaves the slightly awkward final sentence (line 223) "Finally we considered only the address of the corresponding author for the attribution papers to the country of origin trusting in the authorship general criteria." I was not implying at all that authors were misrepresenting their contribution. I would prefer that this paper acknowledge that is not able to capture collaborations, but either way I would prefer to see this sentence deleted.

Additional comments

The changes you need are small but vital to the credibility of your paper.

---

## Round 0.5 · accepted · Accept

Dear Authors,

I would like to compliment with you for the efforts provided in addressing the Reviewers' comments.

Your manuscript is now suitable for publication and can be accepted in its current form.

Best regards and happy New Year 2019

Salvatore Andrea Mastrolia
PeerJ Academic Editor